# Synthesis of Yttria Stabilized Bismuth Oxide by DC Reactive Magnetron Sputtering (RMS) for SOFC Electrolyte

**Xiaolei Ye** [1,2]**, Li Yang** [3,]*****, Huan Luo** [4,]*****, Pierre Bertrand** [5]**, Alain Billard** [1,2] **and Pascal Briois** [1,2]

1   FEMTO-ST Institute (UMR CNRS 6174), Université de Bourgogne Franche-Comté (UBFC),
    Université de Technologie de Belfort Montbéliard (UTBM), Site de Montbéliard, F-90010 Belfort, France
2   USR FC Lab (CNRS FR3539), F-90010 Belfort, France
3   Faculty of Metallurgical and Energy Engineering, Kunming University of Science and Technology,
    Kunming 650093, China
4   School of Materials Science and Engineering, Xi'an University of Science and Technology, Xi'an 710054, China
5   ICB (UMR CNRS 6303), Université de Bourgogne Franche-Comté (UBFC), Université de Technologie de
    Belfort Montbéliard (UTBM), Site de Sévenans, F-90010 Belfort, France
*   Correspondence: 20140005@kust.edu.cn (L.Y.); luohuan@xust.edu.cn (H.L.)

**Abstract:** In this work, the compound of $Bi_{1.5}Y_{0.5}O_3$ was deposited from two metallic targets, respectively, Bi and Y; for a power fixed on the target of Y, the power on the target of Bi was adjusted in order to obtain the desired composition. The effects of atomic ratio of Bi to Y and annealing temperature on the film morphology and crystal structure were investigated. The X-ray diffraction (XRD) results showed different crystal structure as a function of the Y content in the film after annealing at 500 °C for 2 h. The $Bi_{1.5}Y_{0.5}O_3$ was obtained with the atomic ratio of Bi to Y adjusted to 3.1. An impurity phase ($Bi_{1.55}Y_{0.45}O_3$, Rhombohedral) appears in the cubic $Bi_{1.5}Y_{0.5}O_3$ after annealing at 600 °C. The field-emission scanning electron microscopy (FESEM) result showed that the $Bi_{1.5}Y_{0.5}O_3$ film after annealing at 800 °C for 2 h is denser than the as-deposited film, despite the presence of some holes. The ionic transport properties of $Bi_{1.5}Y_{0.5}O_3$ film was measured by electrochemical impedance spectroscopy (EIS), and the conductance activation energy was obtained on this basis. The synthesized $Bi_{1.5}Y_{0.5}O_3$ film with higher ionic conductivity (0.13 S/cm at 650 °C) is suitable for SOFC electrolyte.

**Keywords:** reactive magnetron sputtering (RMS); SOFC; $Bi_{1.5}Y_{0.5}O_3$ film





## 1. Introduction

The solid oxide fuel cell is an efficient energy conversion device that can convert chemical energy into electrical energy through electrochemical reactions [1,2]. However, the development of traditional SOFC technology has been limited due to high cost caused by the high operating temperature (800–1000 °C) [3]. Lowering the operating temperature to introduce inexpensive component materials is an effective way to reduce costs [4]. Nevertheless, compared with high temperature, SOFC exhibits a significant decrease in electrochemical performance due to both the limitation of oxygen ion transport through the electrolyte and oxygen reduction reaction (ORR) at the cathode under medium and low temperature conditions (<750 °C) [5]. In order to compensate for the performance loss, it is necessary to develop some new component materials for SOFC and/or reduce the thickness of the electrolyte. Cubic δ-$Bi_2O_3$ with fluorite structure is perhaps the best oxygen ion conductor among such a category of materials (1 s/cm at 730 °C) [6], and it exhibits both high oxygen ion conductivity and electrocatalytic activity to transform $O_2$ into $O^{2-}$ [7]. As an ionic conductor, the oxygen ion conductivity of cubic δ-$Bi_2O_3$ is 1–2 orders of magnitude higher than that of conventional yttria stabilized zirconia [8–10]. However, cubic δ-$Bi_2O_3$ has a limited operating temperature range for SOFC applications, melting above 823 °C and phase transformation to monoclinic α-phase below 730 °C, resulting in greatly reduced conductivity [4,9]. Fortunately, numerous studies have shown

that the stable temperature of cubic $\delta$-$Bi_2O_3$ can be extended to room temperature by doping with specific elements, mainly trivalent lanthanides, but at the expense of reduced ionic conductivity [11–13]. Unfortunately, bismuth oxide is not thermodynamically stable and is reduced to metallic bismuth under the reducing atmosphere [14,15]. Therefore, even stabilized bismuth oxide is usually combined with YSZ or GDC to form a bilayer electrolyte for SOFC applications [16–19]. Zhang et al. [20] studied $Sm_{0.2}Ce_{0.8}O_{1.90}$ (SDC) and $Bi_{0.75}Y_{0.25}O_{1.5}$ (YSB) electrolytes and showed that interfacial polarization resistance of LSM-YSB cathodes measured with symmetrical cells was supported on SDC than YSB substrates. Bal et al. [21] synthesized 28 mol% $Y_2O_3$ doped $Bi_2O_3$ (YDB), showing a high ionic conductivity of $1.1 \times 10^{-1}$ s/cm by the powder sintering method. Panuh et al. [22] prepared a $Y_{0.25}Bi_{0.75}O_{1.5}/Sm_{0.2}Ce_{0.8}O_{1.9}$ bilayer system with an extremely low total polarization resistance for low-temperature SOFCs.

It has been clearly known that reducing the thickness of the electrolyte from hundreds of micrometers to several micrometers allows the operating temperature to be reduced from 900–1000 °C to 650–800 °C [23]. Although traditional well-studied preparation methods of electrolytes, such as slip casting, screen printing, electrophoretic deposition, plasma spraying, etc., have relative simplicity and low cost, they are not suitable for preparing electrolyte thin films [24]. As a thin-film deposition technique, physical vapour deposition (PVD) is known for its high reliability and reproducibility. The advantage of this method is that a dense and uniform thin film can be achieved, and it also provides a possibility for the deposition process on a large surface area. In addition, the typical PVD deposition process is generally performed at a relatively low temperature (<500 °C), which can avoid the interface reaction between the electrolyte and the electrode caused by high-temperature sintering (>1000 °C).

In this work, a PVD method—direct current reactive magnetron co-sputtering—was used to deposit yttria-doped bismuth oxide (YSB). $Bi_{1.5}Y_{0.5}O_3$ was obtained by adjusting the sputtering parameters. XRD and SEM were used to observe the morphology and phase change of $Bi_{1.5}Y_{0.5}O_3$ film as a function of the annealing temperature. The ionic conductivity of $Bi_{1.5}Y_{0.5}O_3$ film as a function of temperature was measured by EIS, and the conductance activation energy was obtained on this basis. The purpose of this work is to provide a reference for PVD deposition of yttria-stabilized bismuth oxide films.

## 2. Experimental Procedure

### 2.1. Thin Film Deposition

The DC reactive magnetron sputtering technique was used to synthesize yttria-stabilized bismuth oxide films using metallic Bi and Y targets with a purity of 99.9% (diameter = 145 mm and thickness = 6 mm) in a mixture of argon and oxygen. The sputtering device is a 90 L sputtering chamber vacuumed by a turbo molecular pump allowing less than $10^{-5}$ Pa as the base vacuum. The argon and oxygen flow rates are controlled by using Brooks flowmeters, and the total pressure measurement is attributed to an MKS Bratron gauge. Four circular planar sources of magnetron sputtering are employed in the sputtering chamber which cool down with water. The substrates are positioned on a substrate holder at the draw distance of 70 mm parallel to the sources and rotating to ensure a homogenous deposition. The Bi and Y targets mounted on the unbalanced magnetron are powered through a dual pulsed DC generator allowing the power discharge control. Alumina plates, silicon wafers, and glass slides are used as substrate supporting films to measure various properties such as phase, conductivity, and chemical composition. All substrates were cleaned with alcohol and then rinsed with soap and water prior to deposition. The sputtering parameters such as deposition pressure, gas flow, frequency of the power source, power of the power source, and pulse duration ($T_{off}$), etc., are shown in Table 1.

**Table 1.** Sputtering parameters for deposition of yttria-stabilized bismuth oxide films.

|  | **Bi Target** | **Y Target** |
|---|---|---|
| Ar flow rate (sccm) | 20 | |
| $O_2$ flow rate (sccm) | 10 | |
| Total pressure (Pa) | 0.2 | |
| Drawing distance (mm) | 70 | 70 |
| Power (W) | 30–50 | 300 |
| Frequency (kHz) | 50 | 50 |
| $T_{off}$ (µs) | 5 | 5 |

### 2.2. Characterizations

The thickness of films was measured using a profilometer (Altisurf 500, Altimet) allowing an accuracy of 20 nm. The top surface and brittle-fracture cross-section images of films were observed via FESEM (JSM 7800F JEOL). The crystal structure of the films was characterized using a XRD (BRUKER D8, $K_{\alpha1+\alpha2}$ Co). Based on the XRD measurements, the average crystallite size was estimated by the Scherer formula [25]. Electrochemical impedance spectroscopy (EIS) measurements were carried out using Solartron 1260 under ambient air in a temperature range of 400–800 °C. A two-electrode configuration was used to perform the EIS measurement process as shown in Figure 1. The conductivity of the thin films can be calculated by Equations (1) and (2). Each impedance spectrum measurement was performed in the frequency range of 1 MHz–10 Hz, with 11 points per decade.

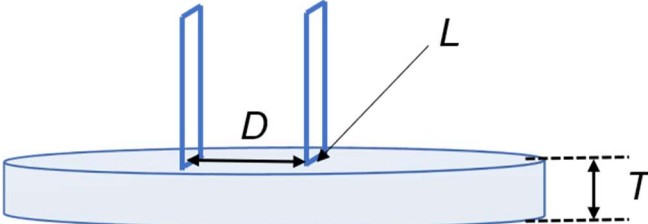

**Figure 1.** Schematic view of the two-electrodes configuration.

$$\sigma = \frac{FG}{R} \tag{1}$$

$$FG = \frac{D}{LT} \tag{2}$$

Here $\sigma$, $FG$, $R$, $D$, $L$, and $T$ are the ionic conductivity, factor of geometry experiment, resistance, electrode distance, electrode width, and thickness of thin film, respectively.

## 3. Result and Discussion

### 3.1. DC Power Affecting on Film Composition

Yttria-doped bismuth oxide (YSB) films with various atomic ratio of Bi/Y in a range of 2.2–6.9 were deposited through DC reactive magnetron sputtering technology. The value of atomic ratio of Bi/Y was obtained by EDX analysis. The atomic ratio was controlled by adjusting the power discharge of the Bi target and fixing the power of the Y target on 300 W. The deposition parameters are given in Table 1. The correlation between the power discharge of the Bi target and atomic ratio of Bi/Y as well as deposition rate are shown in Figure 2. As shown, the relationship between the atomic ratio of Bi/Y, as well as the deposition rate and the power of the Bi target, follows a non-linear trend. This is attributed to the increased sputtering rate of the target surface at high sputtering power. As the power of the Bi target increases, both the atomic ratio of Bi/Y and the deposition rate increase significantly, and the trend becomes more obvious. The films with different atomic ratios of Bi/Y (2.2, 3.1, 4.9, and 6.9) were obtained by adjusting the discharge power of the Bi target.

The as-deposited films were amorphous-nanocrystalline. Thus, a post-annealing treatment at 500 °C under the air was necessary to obtain the crystallized phase. XRD measurement was used to confirm the phase information of the thin films. It can be seen from Figure 3a,b that the films have pure phases corresponding to $Bi_{1.33}Y_{0.67}O_3$ and $Bi_{1.5}Y_{0.5}O_3$ when the atomic ratio of Bi/Y are 2.2 and 3.1, respectively. It is more obvious especially after the annealing treatment at 500 °C. However, the situation changes when Bi/Y is higher, as shown in Figure 3c,d. A YSB main phase is believed to exhibit in Figure 3c,d, while two trace phases, $Bi_{1.55}Y_{0.45}O_3$ and $Bi_2O_3$, are also observed after the annealing treatment at 500 °C. In addition, the films before annealing have a vitreous or even slightly columnar appearance which is accentuated with annealing (Figure 4). This change in morphology is in agreement with X-ray diffraction measurements (Figure 3), which revealed an increase in crystallization with annealing.

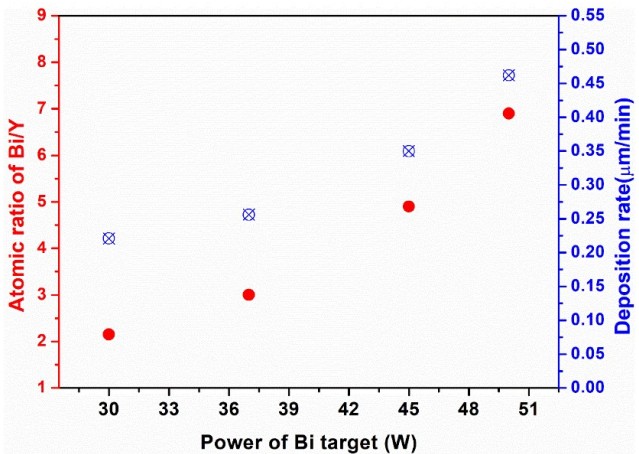

**Figure 2.** The atomic ratio of Bi to Y and the deposition rate of the films as a function of the power of the Bi target.

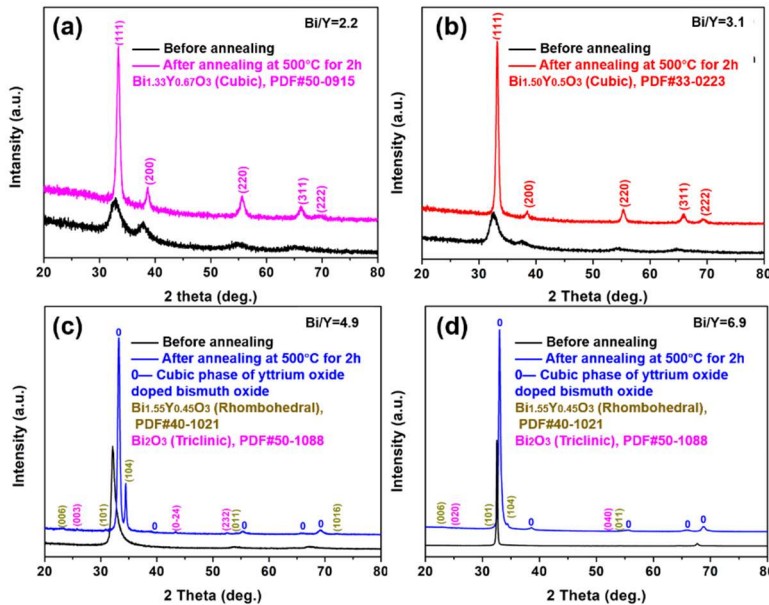

**Figure 3.** XRD characterization of films deposited on glass with different atomic ratios of (**a**) Bi/Y = 2.2, (**b**) Bi/Y = 3.1, (**c**) Bi/Y = 4.9, (**d**) Bi/Y = 6.9.

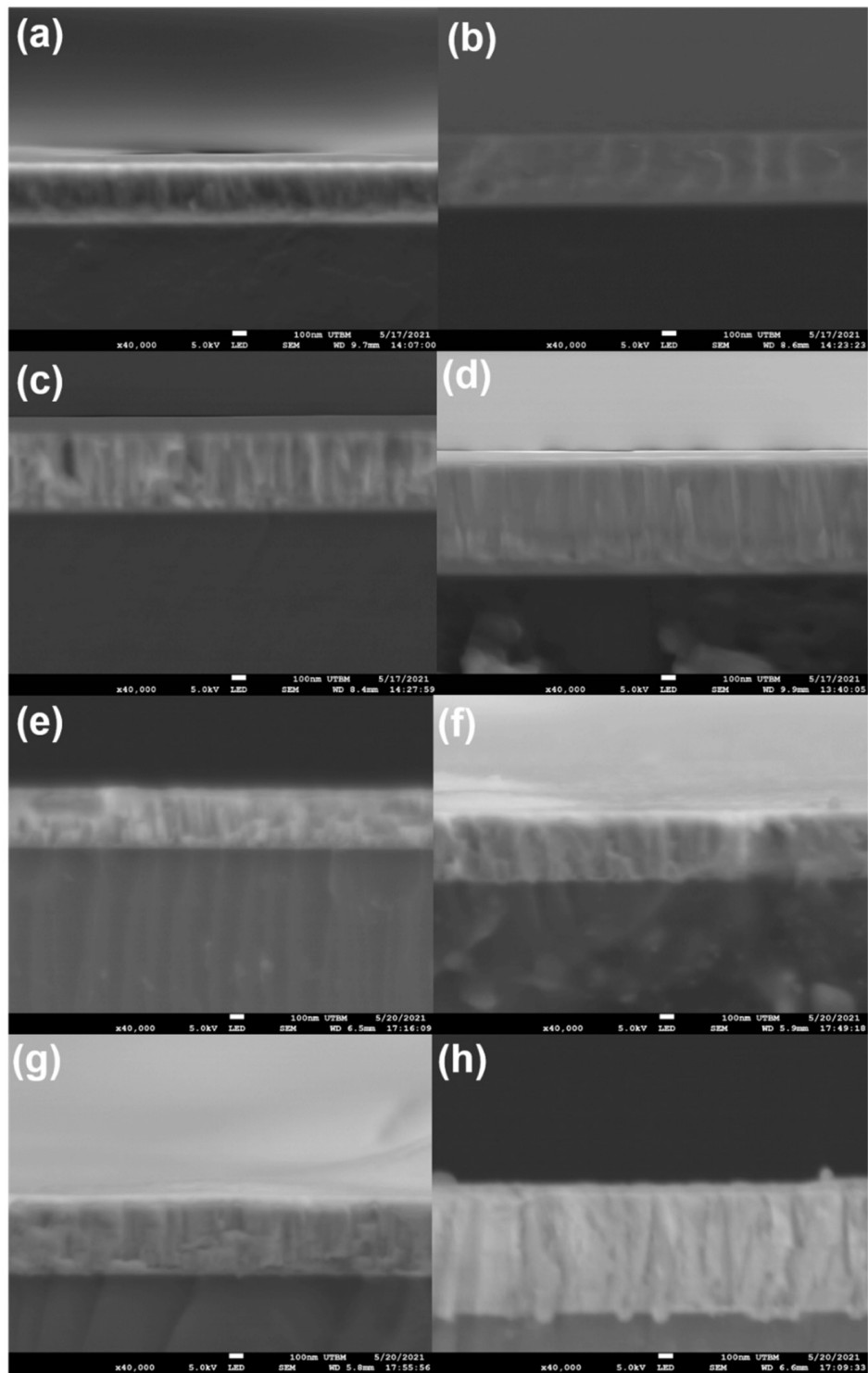

**Figure 4.** FESEM characterization of the brittle fracture cross-section of the films deposited on silicon wafer with different atomic ratios of as deposited (**a**) Bi/Y = 2.2, (**b**) Bi/Y = 3.1, (**c**) Bi/Y = 4.9, (**d**) Bi/Y = 6.9 and annealed at 500 °C (**e**) Bi/Y = 2.2, (**f**) Bi/Y = 3.1, (**g**) Bi/Y = 4.9, (**h**) Bi/Y = 6.9.

### 3.2. Deposition of $Bi_{1.5}Y_{0.5}O_3$ Film via DC RMS

As mentioned above, bismuth oxide doped with yttrium oxide exhibits good ionic conductivity. Therefore, in this work, $Bi_{1.5}Y_{0.5}O_3$ became the research object. According to Section 3.1, the crystal structure of $Bi_{1.5}Y_{0.5}O_3$ was obtained when the atomic ratio of Bi to Y was about 3.1 after annealing at 500 °C. The $Bi_{1.5}Y_{0.5}O_3$ thin films with a thickness of

1.72 µm were deposited on the alumina pellet and glass by controlling the deposition time. The thin film deposited on glass after annealing at 500 °C for 2 h was characterized by XRD, and profile matching was performed by the Rietveld method (Figure 5). δ-Cubic $Bi_{1.5}Y_{0.5}O_3$ was further confirmed and its cell parameters were obtained. The thin films deposited on the alumina wafer were used to observe the phase transition, crystallite size, and the change of the morphology with the annealing temperature range of 400–800 °C (Figure 6 and Table 2). As seen from Figure 6, an impurity phase ($Bi_{1.55}Y_{0.45}O_3$, Rhombohedral) appears after annealing at 600 °C. As the annealing temperature further increases, the impurity phase gradually decreases. At the same time, as the annealing temperature increases, the FWHM of XRD gradually decreases, which means that the crystallite size of the thin films increases (Table 2). This is consistent with the morphology observed by FESEM (Figures 7 and 8). In addition, Figures 7 and 8 show that the thin films are denser with increasing annealing temperature. However, holes appear on the surface of the thin films over 600 °C. Further, the composition evolution of the films with annealing temperature was investigated by EDS. The EDX results in Figure 9 show that the content of oxygen atoms decreases with increasing annealing temperature, which may be attributed to the escape of absorbed elemental oxygen during the magnetron sputtering process. This may be one of the reasons for the cracks and holes in the film. The atomic ratio of B to Y of the films decreased from 3.19, as deposited, to 2.95 annealing at 800 °C, which should be attributed to the evaporation of Bi. This may be one of the reasons that led to the formation of holes in the film at high temperature. The derision reduction of atomic ratio of B to Y may also be the reason for the reduction of the impurity phase ($Bi_{1.55}Y_{0.45}O_3$, Rhombohedral). In addition, the formation of cracks and holes resulted in an increase in the Al content in the EDX results since the thin films were deposited on an alumina pellet.

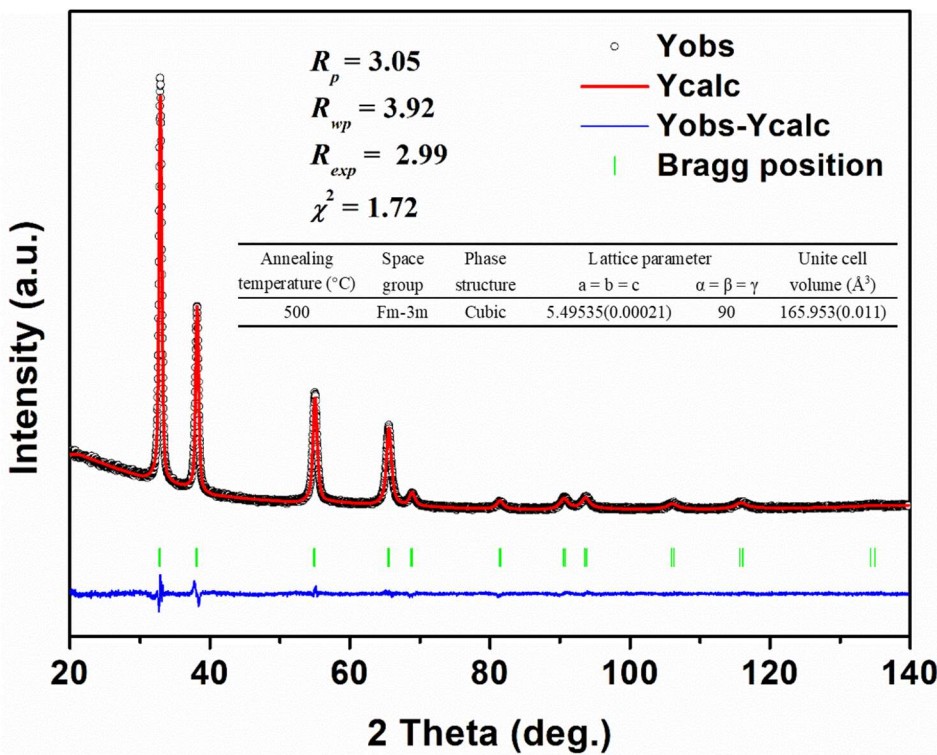

**Figure 5.** The XRD characterization and profile matching of the $Bi_{1.5}Y_{0.5}O_3$ film of 1.72 µm deposited on glass after annealing at 500 °C for 2 h.

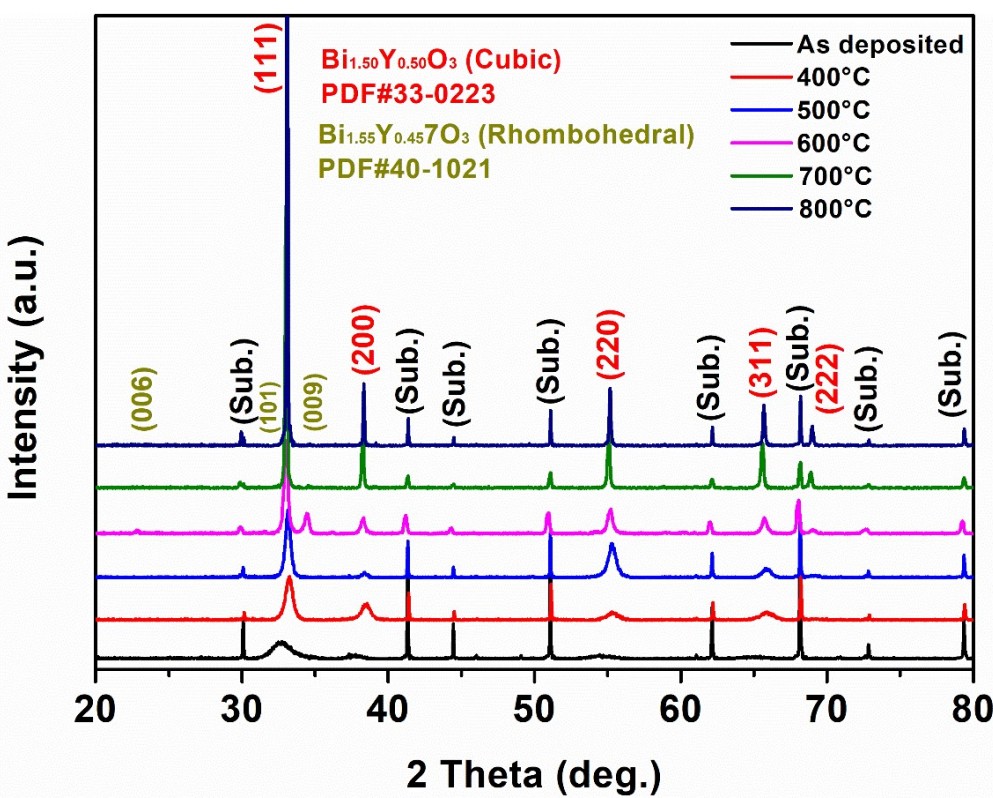

**Figure 6.** The XRD characterization of the $Bi_{1.5}Y_{0.5}O_3$ films varies with the different annealing temperature.

**Table 2.** The crystallite size of $Bi_{1.5}Y_{0.5}O_3$ films varies with the different annealing temperature from XRD characterization.

| Annealing Temperature (°C) | FWHM (Å) | Crystallite Size (Å) |
| :---: | :---: | :---: |
| As deposited | 1.550 | 62 |
| 400 | 0.619 | 158 |
| 500 | 0.442 | 223 |
| 600 | 0.329 | 307 |
| 700 | 0.179 | 647 |
| 800 | 0.132 | >1000 |

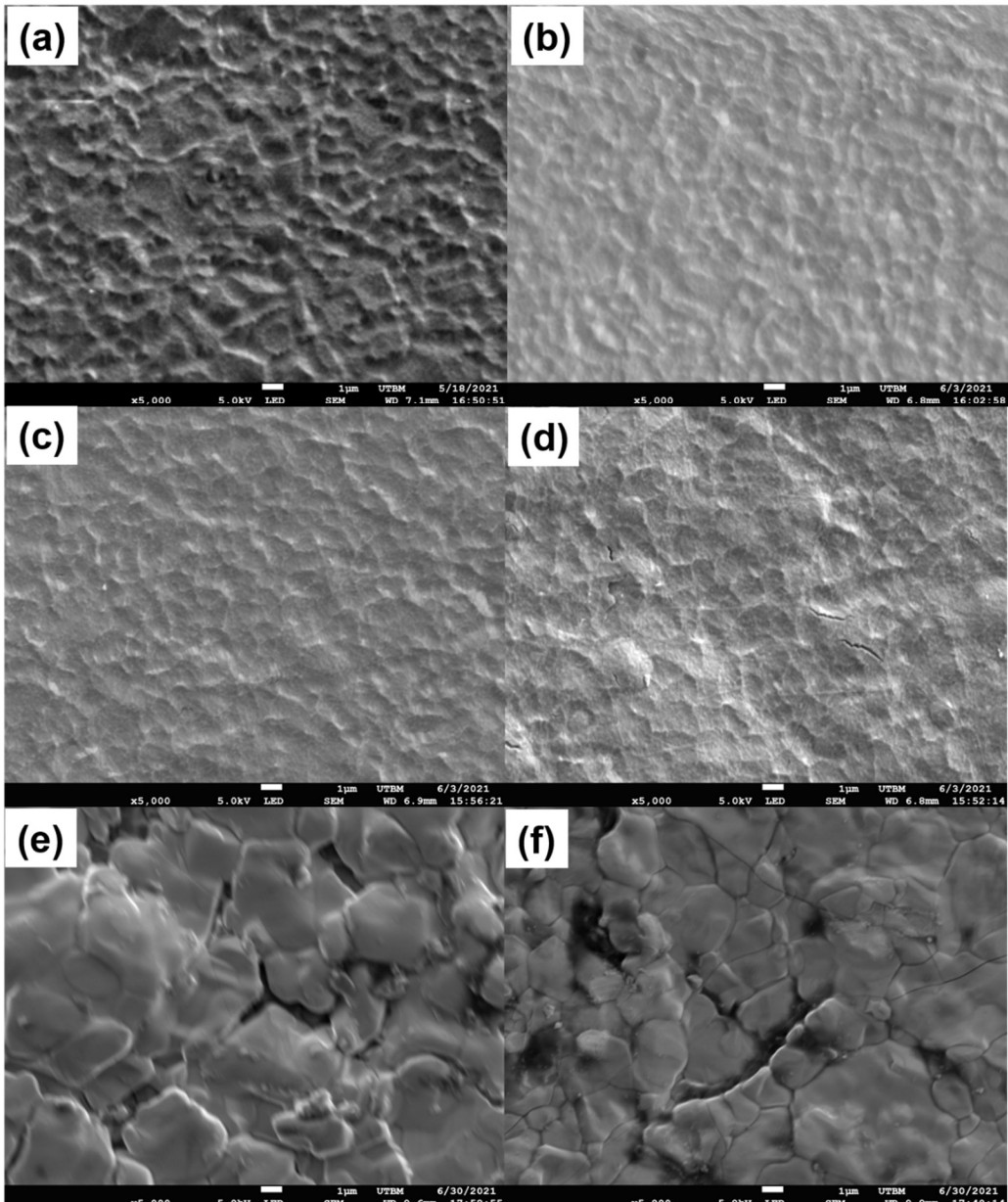

**Figure 7.** FESEM characterization of the top surface of the $Bi_{1.5}Y_{0.5}O_3$ films deposited on alumina plate varies with the different annealing temperature of (**a**) as deposited, (**b**) 400 °C, (**c**) 500 °C, (**d**) 600 °C, (**e**) 700 °C, (**f**) 800 °C.

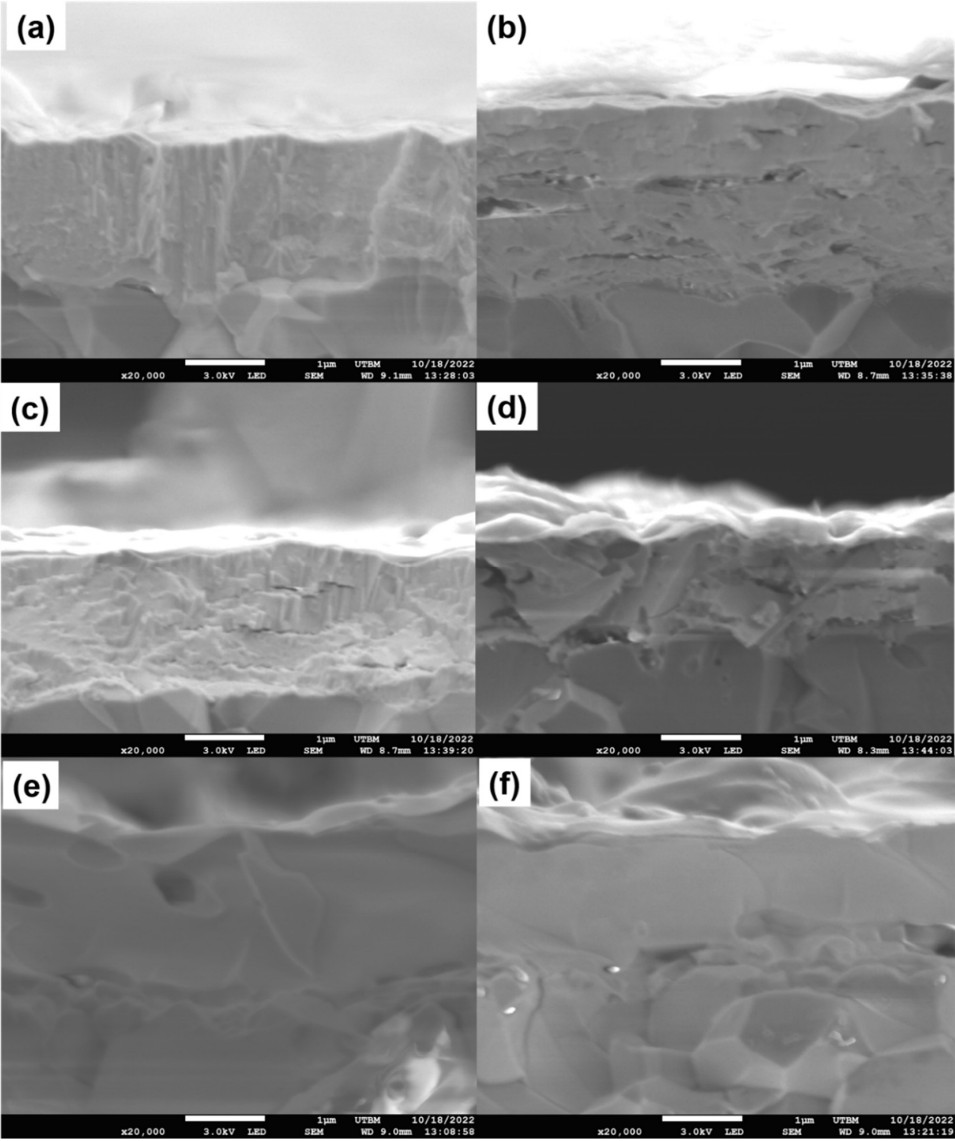

**Figure 8.** FESEM characterizes of the brittle fracture cross-section of the Bi$_{1.5}$Y$_{0.5}$O$_3$ films deposited on alumina plate varies with the different annealing temperature of (**a**) as deposited, (**b**) 400 °C, (**c**) 500 °C, (**d**) 600 °C, (**e**) 700 °C, (**f**) 800 °C.

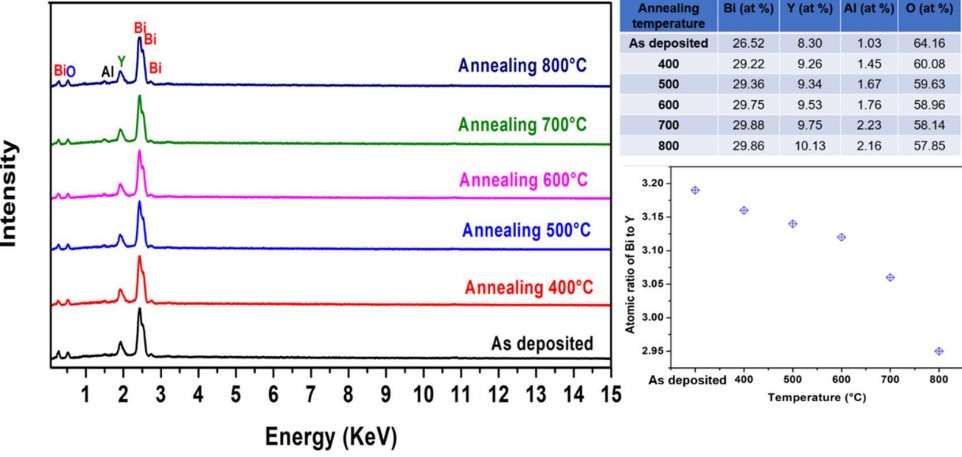

| Annealing temperature | Bi (at %) | Y (at %) | Al (at %) | O (at %) |
|---|---|---|---|---|
| As deposited | 26.52 | 8.30 | 1.03 | 64.16 |
| 400 | 29.22 | 9.26 | 1.45 | 60.08 |
| 500 | 29.36 | 9.34 | 1.67 | 59.63 |
| 600 | 29.75 | 9.53 | 1.76 | 58.96 |
| 700 | 29.88 | 9.75 | 2.23 | 58.14 |
| 800 | 29.86 | 10.13 | 2.16 | 57.85 |

**Figure 9.** The EDX results of films deposited on alumina plate as function as temperature.

### 3.3. The Electrochemical Properties of the Bi$_{1.5}$Y$_{0.5}$O$_3$ Film

Ionic conductivity is an important indicator for SOFC electrolytes. Generally, the ionic conductivity of the electrolyte is required to be no less than 0.1 S/cm at the operating temperature [26]. Here, the electrochemical impedance spectroscopy of Bi$_{1.5}$Y$_{0.5}$O$_3$ thin films deposited on the alumina plate was measured following the test method shown in Figure 1, and the corresponding conductivity was calculated by Equations (1) and (2). From Figure 10, the EIS results show that one main semi-circle is attributed to the thin film before 600 °C. However, over 600 °C, there is a change in the impedance spectrum in the low frequency region, which results from the electrode effects [27,28]. The possible reason for this phenomenon is the change in the surface morphology of the thin films. Here, ignoring the electrode effects, an equivalent circuit (resistor parallel constant phase element) was used to fit the semi-circular arc to obtain the corresponding resistance and calculate the conductivity (Table 3). From Table 3, the ionic conductivity of the thin film increases with increasing temperature. The thin film exhibits good ionic conductivity at temperatures higher than 600 °C. Furthermore, the Arrhenius conductivity plot for Bi$_{1.5}$0Y$_{0.5}$O$_3$ is shown in Figure 11. As observed from Figure 11, the activation energy in the lower (<600 °C) and higher (>600 °C) temperature range shows a clear difference. The conductivity activation energies are 1.60 eV and 0.81 eV in the lower and higher temperature range, respectively. This indicates that the oxide ion conduction of Bi$_{1.5}$Y$_{0.5}$O$_3$ thin films may involve two temperature-dependent activation processes [27].

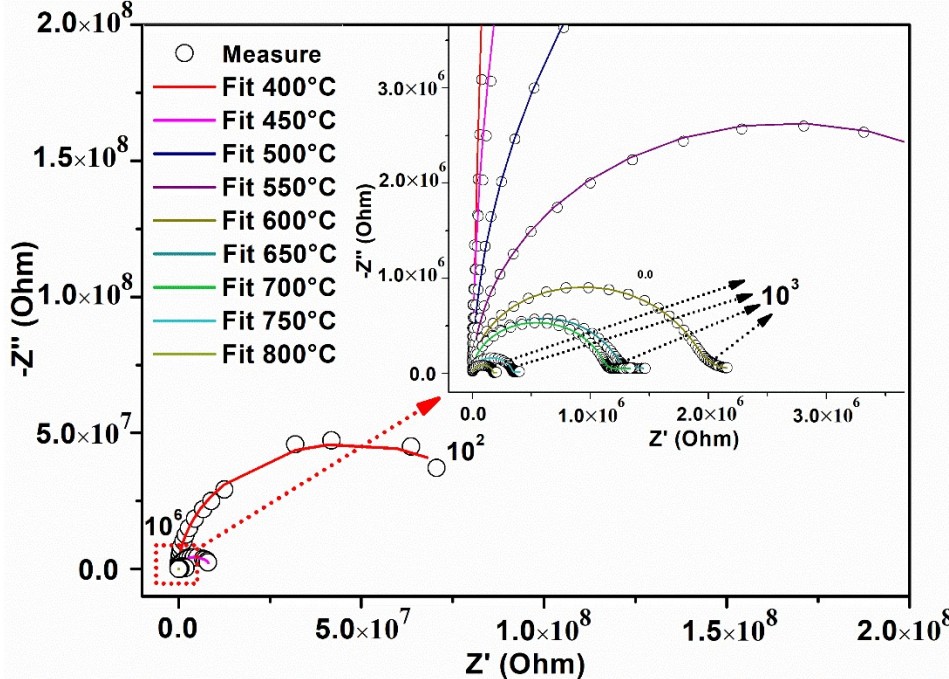

**Figure 10.** The electrochemical impedance spectroscopy of Bi$_{1.5}$Y$_{0.5}$O$_3$ film with a thickness of 1.72 μm varies with the measured temperature range from 400–800 °C.

**Table 3.** The conductivity and resistance of $Bi_{1.5}Y_{0.5}O_3$ film with a thickness of 1.72 µm.

| | $R_e$ | $\sigma$ |
|---|---|---|
| 400 °C | $8.55 \times 10^7$ | $1.36 \times 10^{-4}$ |
| 450 °C | $8.96 \times 10^6$ | $1.30 \times 10^{-3}$ |
| 500 °C | $1.99 \times 10^6$ | $5.83 \times 10^{-3}$ |
| 550 °C | $5.43 \times 10^5$ | $2.14 \times 10^{-2}$ |
| 600 °C | $1.29 \times 10^5$ | $8.98 \times 10^{-2}$, $6.35 \times 10^{-2}$ [29], $4.38 \times 10^{-2}$ [30] |
| 650 °C | $8.90 \times 10^4$ | $1.31 \times 10^{-1}$, $1.1 \times 10^{-1}$ [21], $1.41.1 \times 10^{-1}$ [31] |
| 700 °C | $6.80 \times 10^4$ | $1.71 \times 10^{-1}$ |
| 750 °C | $3.12 \times 10^4$ | $3.73 \times 10^{-1}$ |
| 800 °C | $1.71 \times 10^4$ | $6.81 \times 10^{-1}$ |

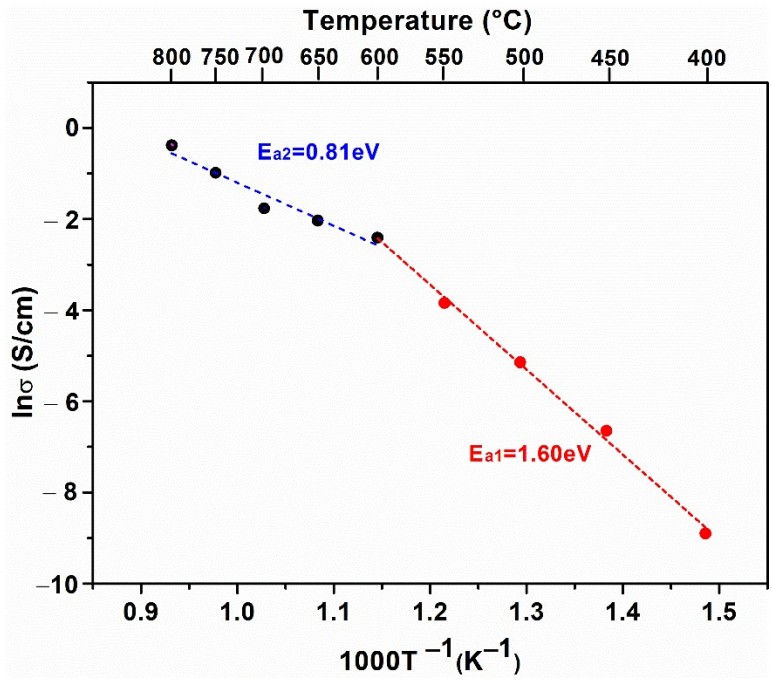

**Figure 11.** The Arrhenius conductivity plots of $Bi_{1.5}Y_{0.5}O_3$ film.

## 4. Conclusions

It has been confirmed that the cubic $Bi_{1.5}Y_{0.5}O_3$ film was deposited by reactive magnetron sputtering. The $Bi_{1.5}Y_{0.5}O_3$ film is thermally stable in the temperature range of 400–800 °C, and a small amount of impurity phase ($Bi_{1.55}Y_{0.45}O_3$, Rhombohedral) is only at 600 °C. At the same time, this film also exhibits higher ionic conductivity (>0.1 S/cm) in the medium temperature range (600–800 °C), which meets the ionic conductivity requirements of the SOFC electrolyte. In addition, the oxygen ion conduction process of the film may involve two different temperature-dependent activation mechanisms because of different activation energies in the lower and higher temperature range, respectively.

**Author Contributions:** Conceptualization, X.Y. and A.B.; Data curation, X.Y.; Formal analysis, L.Y.; Investigation, X.Y.; Methodology, X.Y.; Resources, P.B. (Pascal Briois); Visualization, H.L. and P.B. (Pierre Bertrand); Writing – original draft, X.Y.; Writing—review & editing, P.B. (Pascal Briois). All authors have read and agreed to the published version of the manuscript.

**Funding:** China Scholarship Council (No. 201808530576), Natural Science Basic Research Plan in Shaanxi Province of China (2022JQ-551).

**Acknowledgments:** The authors thank the China Scholarship Council, Pays de Montbéliard Agglomeration and the Natural Science Basic Research Plan in Shaanxi Province of China.

**Conflicts of Interest:** The authors declare no conflict of interest.

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
