# Peer review of "Synthesis of Yttria Stabilized Bismuth Oxide by DC Reactive Magnetron Sputtering (RMS) for SOFC Electrolyte"

_crystals, doi:10.3390/cryst12111585_

Round 1
Reviewer 1 Report
Dear authors
I read the article with great interest. I believe it is worthy of publication, but I also think it first requires discussion and consideration of several issues. I think it would be worthwhile for the authors to read my suggestions and consider making corrections, or at least help me understand some of the points raised. Here is a list of my concerns:
1. Abstract
I got the impression that the abstract puts too much emphasis on the background of the state of the issue, and does not treat enough about the results of the realized experiment.
2. Experimental procedure, Thin films deposition
The description of the process vaguely explains its conditions. It is not clear from it in what arrangement the magnetrons were used. Was it a confocal arrangement? How was the power supply to the magnetrons executed? Were they powered from a shared power supply alternating in a cycle or from two power supplies simultaneously?
3. Table 1
What exactly does the Toff parameter refer to? The text does not explain it.
4. Results and discussion, DC power affecting on film composition
If the purpose of the article was to enrich bismuth oxide with yttrium oxide why did the authors change the sputtering power of the bismuth target and not yttrium? Were there any technological obstacles to realize the process with different sputtering power of the yttrium target?
5. Page 3, lines 110-111
“As shown, the relationship between the atomic ratio of Bi/Y, as well as the deposition rate and the power of Bi target follows a non-linear trend.” This can be easily explained by an increase in the sputtering rate of oxide phases on the target surface at high sputtering power.
6. Page 3, lines 114-115
“The as-deposited films were amorphous.” Cannot agree with this statement. Of course, the boundary between the nanocrystalline and amorphous phases is not sharp. Nevertheless, the diffractograms rather indicate that it is not an amorphous material, as diffraction signals on crystallographic planes are visible on it. It should be classified as amorphous-nanocrystalline.
7. Fig. 3
The subscripts in the phase descriptions of yttrium and bismuth oxide are barely visible
8. Fig. 4, 7 and 8
The thickness scale in the images is impossible to see. Do the authors have the images from Fig. 4 in better resolution? The layer structure in the images is not well distinguished.
9. Discussion
Are the authors able to explain the mechanism of the formation of triclinic Bi2O3 and rhombohedral Bi-Y-O phases instead of cubic Bi-Y-O as a result of the technological variants used? The text does not propose any mechanism.
10. Discussion
In the discussion, can the authors relate their results to those of similar layers reported in the literature? This would greatly increase the relevance of the paper.
11. References
The list of references is quite deficient for a full - length article. In my opinion, it is worth enriching it with more positions. I got the impression that many statements in the text are not supported by any literature, especially in the Introduction section.
Reviewer 2 Report
In this contribution, (Bi,Y)2O3 thin films are prepared by dc magnetron sputtering on different substrates and their structure, microstructure and electrical properties are investigated. In general, the work is of interest for the SOFC community; however, before considering for publication several issues need to be addressed.
1. The first paragraph in the Abstract is similar to the introduction section and could be omitted or rewritten.
2. Bi2O3 based materials are not redox stable under the reducing atmosphere of the anode and therefore they are not suitable as electrolyte in SOFCs. However, they are of great interest for application as active functional layers to improve the efficiency of the air electrodes in SOFC. I think that this point should be commented in the introduction section, including adequate references, e.g. 10.1016/j.ijhydene.2021.02.217 , 10.1002/admi.202200702, 10.1039/d1ta07308g, 10.1016/j.ceramint.2021.10.076, 10.1016/j.ijhydene.2018.03.168, 10.3390/applnano1010003
3. It is recommendable to determine the cation composition of the films at different annealing temperature to rule out possible Bi evaporation, which could explain the crack formation.
4. The values of conductivity need to be compared with those reported in the literature for thin films and thick pellets.
5. The list of references (11) should be improved. Numerous relevant papers have been published in recent years. Some key, important or/and latest research results in this field should be mentioned and cited in the introduction section to provide a solid background and progress to the readers regarding the current state-of-knowledge on this topic. Therefore, I strongly require to rewrite this part and then update the citations.
Round 2
Reviewer 2 Report
The manuscript has been revised according to the suggestions and comments of the reviewers and can be accepted.